# Analysis of Self-Perceived Physical Fitness of Physical Education Students in Public Schools in Extremadura (Spain)

**DOI:** 10.3390/children10030604

**Published:** 2023-03-22

**Authors:** Carmen Galán-Arroyo, David Manuel Mendoza-Muñoz, Jorge Pérez-Gómez, Claudio Hernández-Mosqueira, Jorge Rojo-Ramos

**Affiliations:** 1Physical and Health Literacy and Health-Related Quality of Life (PHYQoL), Faculty of Sport Science, University of Extremadura, 10003 Cáceres, Spain; 2Physical Activity for Education, Performance and Health, Faculty of Sport Sciences, University of Extremadura, 10003 Cáceres, Spainjorgerr@unex.es (J.R.-R.); 3Health Economy Motricity and Education (HEME), Faculty of Sport Science, University of Extremadura, 10003 Cáceres, Spain; 4Departamento de Educación Física, Deportes y Recreación, Universidad de La Frontera, Temuco 4780000, Chile

**Keywords:** physical fitness, childhood, physical education, self-perceived physical fitness

## Abstract

Adolescence is a stage of crucial physiological and psychological changes within the individual’s life cycle, where fitness work is important. With self-perception being crucial in relation to adolescent health and well-being, a positive perception of fitness is directly related to increased practice or higher level of physical activity (PA). Thus, the aims were: (1) to analyze, with the Visual Analogue Fitness Perception Scale for Adolescents (FP VAS A), the self-perceived physical fitness (PF) of high school students, (2) to investigate if there are differences according to sex and school location, and (3) to study the correlations between the items of the FP VAS A with age and body mass index (BMI). For this purpose, a cross-sectional study was designed with a total of 961 participants, 48.8% boys and 51.2% girls in secondary education, where 31.9% studied in rural schools and 68.1% in urban schools. The FP VAS A scale was used to assess self-reported PF. Regarding the results, there were statistically significant differences between sexes (*p* < 0.001), with boys showing higher scores than girls in all the items of the FP VAS A scale, with the exception of global flexibility. Inverse, mean and significant correlations were established between BMI and self-perceived overall PF (r = −0.202; *p* < 0.001), cardiorespiratory endurance (r = −0.226; *p* < 0.001) and movement speed (r = −0.268; *p* < 0.001). Between age and self-perceived cardiorespiratory endurance (r = −0.138; *p* < 0.001) an inverse, mean and significant correlation was also observed. In conclusion, boys showed a better self-perception of PF than girls for all physical abilities, with the exception of flexibility. School location was not shown to influence students’ self-perceived PF. In addition, most of the self-perceived PF abilities for overall fitness correlated inversely with BMI.

## 1. Introduction

Physical fitness (PF) is considered as the capacity of the set of physical attributes available to the organism to perform different types of physical activity (PA) in a controlled and efficient manner, without generating excessive fatigue [1,2]. The work on PF is essential for its improvement or maintenance, since it is an indicator of great relevance for the development, growth and health of children and adolescents [3,4], and PF is even associated with cognitive functions and academic performances of students [5,6,7]. Therefore, childhood and adolescence, as stages where a large number of crucial physiological and psychological changes take place in the life cycle of the individual, are crucial in the work of PF since at these ages, healthy lifestyles and behaviors are established that will have an impact on the state of health and quality of life of the individual in later life [8,9].

The relationship between PF and health and the prevention of pathologies is very close, and is considered one of the most important markers of health [4,10]. An adequate level of PF during adolescence can improve cardiovascular function and may help protect these individuals from future cardiovascular disease in later life [11,12]. In this sense, working on PF from an early age is crucial, since these are the initial stages of atherosclerotic cardiovascular disease, where it usually manifests clinically and later appears in adulthood [13]. In addition, low levels of PF during adolescence in some physical capacities, such as cardiorespiratory fitness and strength, is associated with an increased risk of type 2 diabetes [14], myocardial infarction [15] and premature mortality [16] during adulthood. Conversely, high levels of fitness during these early stages are inversely related to total and abdominal obesity [17] and, specifically, strength, speed and aerobic capacity were shown to be significantly associated with adiposity and adiposity growth in childhood and puberty. Thus, improving fitness levels during these stages is paramount to counteracting future obesity [17,18]. In this sense, public health agencies have a great interest in assessing PF and investigating various methods to obtain an effective assessment in order to develop relevant interventions and to detect early the manifestation of some pathologies [4,19].

There are different means and methods to evaluate the PF of individuals, including laboratory tests together with the associated use of specific devices and instruments, which is the most objective way to obtain accurate measurements of the different parameters that make up PF [20]. However, these methods are oriented to an individual level, and are more expensive and require more time and qualified personnel to be performed correctly; therefore, in the educational field, this method would not be viable for all students in a physical education class to know and manage their PF values [21]. For this reason, other alternatives such as field tests have proven to be valid and reliable for these situations, consisting of cheap, minimal and easy to use materials, where several participants can perform the tests simultaneously, such as the test batteries to assess PF in young people and adults, including the European Physical Fitness Battery (EUROFIT) [22], Physical Activity and Health Battery for Adults (AFISAL) [23] and the Assessing Levels of Physical Activity Battery (ALPHA-Fitness) [24].

However, these assessments still require a large amount of time to obtain the results of all the individuals in a group and, in this sense, another alternative could be to use self-reported fitness assessments, which could solve this problem by having the whole class take the survey simultaneously, requiring only a few minutes to complete [25,26]. These instruments, such as the International Fitness Scale (IFIS) [26], the Visual Analogue Fitness Perception Scale for Adolescents (FP VAS A) [25] (the scale used in the present research) and the Delignières et al. [27] self-perception PF questionnaire, slightly modified by Jürimäe et al. [28], provide a subjective assessment of fitness for each of the component abilities, providing valuable information for adolescents to be aware of their deficiencies and thus be able to address them.

Self-perception of PF is of great significance in adolescence in relation to the health and well-being of adolescents themselves, since, as has been proven in various studies, a positive perception of PF is directly related to a greater practice or higher level of PA [29,30]. This aspect can influence adolescents into adopting active and healthy lifestyle habits to improve the perception of their PF, as well as increase their self-esteem and confidence, helping them to face challenges and develop in other areas of their lives [30,31].

Currently, there are some studies that have compared in a secondary way the self-perception of PF between boys and girls, with boys obtaining a better self-perception of PF in all physical abilities except flexibility [30,32,33]. However, there are hardly any studies that compare the self-perception of PF and the location of the center (rural or urban), and there is controversy because there are previous studies where FP was measured from different tests and the rural population had a better FP [34,35], but recent articles show that in cities since there are now more extracurricular activities, they do more PA and might have better FP [36]. It could also be influenced by the socio-economic status of the participants. There is not much evidence regarding the relationship between age and body mass index (BMI) with self-perception of PF, and studies are limited to the level of PF and not to self-perception [37,38,39]. Therefore, in the present investigation, we intend to analyze, through the FP VAS A items, the self-reported PF of secondary school students and to investigate whether there are differences according to sex and center location. On the other hand, the correlations between the items of the FP VAS A with age and BMI will also be studied.

## 2. Materials and Methods

### 2.1. Participants

The sample size was selected following the non-probability sampling method based on convenience sampling [40]. A total of 961 participants in secondary education were assessed.

The inclusion criteria for participants were: (a) have informed parental consent; (b) is a student in the area of physical education in public schools in Extremadura at the secondary education level (from twelve to eighteen years of age).

The study was conducted in accordance with the ethical provisions of the Declaration of Helsinki and the protocol was approved by the Bioethics Committee of the University of Extremadura (Registration Code 71/2022).

Table 1 shows the sociodemographic characterization of the sample. Of the total sample (n = 961), 48.8% were boys and 51.2% were girls, so the sample can be considered to be gender-balanced. Regarding school location, 31.9% studied in rural schools and 68.1% studied in urban schools. The schools located in towns with less than 20,000 inhabitants were considered as rural schools and those with more than 20,000 inhabitants as urban schools, following the criteria established by the Cáceres Provincial Council [41]. The mean age was 14.71 years (SD = 1.58) and the mean BMI (kg weight/height in meters^2^).

### 2.2. Procedure

Based on the directory of public schools in Extremadura provided by the Ministry of Education and Employment of the Regional Government of Extremadura, contact details were selected for all those teaching at the Compulsory Secondary Education (CSE) (ages 12 to 16) and Baccalaureate (ages 16 to 18) levels.

An e-mail was sent to all the selected centers addressed to the physical education teachers, informing them about the object of the study, a model of the instrument and parental informed consent forms.

On the agreed day, a researcher went to the school and, after verifying that the parents or guardians of the participants who were in the physical education class had signed the informed consent form, proceeded to provide each student with a tablet with the URL link to the questionnaire. The questionnaire was elaborated with the digital application Google Forms and the researcher read aloud each item of the questionnaire to ensure that the participants had understood the questions. It was decided to use an e-questionnaire to more easily store all the responses in the same database, saving time and costs.

The average time taken to complete the questionnaire was 10 min. All data were collected anonymously between September and December 2022.

### 2.3. Instruments

Sociodemographic data: A questionnaire was prepared with six questions aimed at characterizing the sample based on sex, age, height, weight, grade and location of the center.

Visual Analogue Fitness Perception Scale for Adolescents (FP VAS A): To assess self-reported PF in adolescents, the Visual Analogue Fitness Perception Scale for Adolescents was used [25]. The scale is composed of five items (general physical condition, cardiorespiratory fitness, muscular strength, speed–agility and flexibility). The instrument uses a Likert scale of 1–10 with 1: very poor level and 10: excellent level. The authors reported a reliability value of the instrument as a Cronbach’s alpha coefficient of 0.860. In our study, the reliability of the instrument was calculated from Cronbach’s alpha statistic and a value of 0.77 was obtained. This can be considered a satisfactory value according to Nunnally et al. [42].

### 2.4. Statistical Analysis

First, to determine the type of statistical tests to be used, the distribution of the data was explored to see if the assumption of normality was met using the Kolmogorov–Smirnov test. This assumption was not met, so it was decided to use nonparametric statistical tests.

To analyze the differences between the scores for each of the variables studied, according to sex or type of center, the Mann–Whitney U test was used. A significance level of *p* < 0.05 was established.

To determine the degree of relationship between each of the variables and age or BMI, the Spearman’s Rho test was used. For the interpretation of this statistic, we took into account the range established by Mondragón Barrera [43], who defined that coefficients between 0.01 and 0.10 indicate the existence of a low correlation, values between 0.11 and 0.50 imply a medium degree of correlation, values from 0.51 to 0.75 indicate a strong correlation, from 0.76 to 0.90 indicate a high correlation, and above 0.91 the correlation is perfect.

To calculate the effect size of sex or center location for each of the variables, Hedges’ g was used. A value below 0.20 indicates no effect, values between 0.21 and 0.49 indicate a small effect, values between 0.50 and 0.79 indicate a moderate effect, and values above 0.80 indicate a strong effect [44].

Finally, Cronbach’s alpha was used to determine the reliability of the instrument. To interpret the values reported, the guidelines established by Nunnally et al. were chosen, which state that values below 0.70 would correspond to low reliability, values between 0.71 and 0.90 would correspond to satisfactory reliability and values above 0.91 would correspond to excellent reliability [42].

The data are presented as a number and percentage for sociodemographic variables and as mean (M) and standard deviation (SD) for scores obtained in each of the variables of the FP VAS A instrument. The software used for data analysis was the Statistical Package of Social Science, version 23 for MAC.

## 3. Results

Table 2 shows the descriptive data (from the mean and standard deviation) and the differences for each of the items that make up the FP VAS A scale according to gender and center location.

Boys obtained higher scores and, therefore, a better self-perception of PF than girls, in all items (*p* < 0.001) except in item 5 “My overall flexibility is” (*p* < 0.001), where girls showed higher scores and showed a better self-perception of flexibility. These differences were statistically significant in all items. Boys also obtained a higher overall score on the FP VAS A scale (*p* < 0.001) than girls, with these differences being statistically significant.

With respect to the location of the center, no statistically significant differences were obtained in any item of the FP VAS A scale. However, rural center students showed slightly higher scores than urban center students on most of the items and on the overall FP VAS A scale score. Students from urban centers only obtained better scores on item 5 “My overall flexibility is”, with the scores of students from both types of centers coinciding on item 4 “My movement speed is”.

To analyze the relationship between each of the items of the FP VAS A scale with age or BMI (Table 3), the Spearman’s Rho test was used. An inverse, low and significant correlation was obtained between the items 3 “My overall muscle strength is” (*p* < 0.035), 4 “My movement speed is” (*p* < 0.009) and age. Furthermore, for item 2 “My cardiorespiratory endurance” (*p* < 0.001) the correlation with age was inverse, medium and significant.

BMI obtained an inverse, medium and significant correlation (*p* < 0.001) with overall physical fitness, with cardiorespiratory endurance capacity and with movement speed. For item 5 “My overall flexibility is” (*p* < 0.033), the correlation with BMI was inverse, low and significant.

Globally, the FP VAS A scale had an inverse, low and significant correlation with age and inverse, medium and significant correlation with BMI.

## 4. Discussion

One of the objectives of the present investigation was to analyze the differences in scores obtained in the self-perception of PF as a function of sex. The results obtained are supported by other studies [25,26,28,30,45,46,47], where boys have better self-perceived fitness for all physical abilities and for general fitness, with the exception of flexibility, where girls showed significantly higher self-perceived fitness scores than boys in our research. In other studies, the self-perception of flexibility was better in boys than in girls and, moreover, the differences obtained were not significant [25,30,47], which is clearly contrary to the results of our study. Jürimäe et al.’s study analyzed how the self-perception of PF evolves throughout adolescence, and it was observed how boys from the ages of 14–15 years progressively decreased their self-perception of flexibility over the years, with girls showing a better self-perception of flexibility at the age of 16–17 years [28]. In relation to this, a large part of the participants were in the age range of 15 to 18 years, so this greater self-perception of flexibility by girls could be associated with the belief that girls are more flexible and boys are more rigid. However, no specific age was established where flexibility begins to decrease without training in boys, and this decrease depends on various factors, such as genetics, nutrition, lifestyle and level of PA [48,49]. Even so, it is common for flexibility to begin to decrease after puberty if a training routine is not maintained [50].

As observed in this research, boys have a more positive perception of their PF than girls, a fact that may be influenced by irregular PA practice or by a low level of PF of girls compared to boys, since it has been proven in previous research that the self-perception of PF has a direct correlation with the level of PA [30,51] and with the level of PF [25,26]. These differences in the self-perception of PF could be related to the physical self-concept, on how the adolescent perceives himself, with boys perceiving themselves better in the five physical competencies related to physical self-perception established by Fox et al.: PF, attractive body, sports competence, physical strength and self-confidence [52]. In this sense, in the present research, girls showed worse self-perception in strength ability, an ability typically associated with boys ahead of girls [53]. In the study by Crocker et al. a weak association was demonstrated for adolescent girls between perception of physical strength and physical self-esteem [54], and other studies claim that boys experience greater perception of physical strength because they perceive a stronger physical self-perception [33,55,56]. Another possible justification for this lower self-perception of PF by girls could be related to the appearance of the signs of puberty and its physiological modifications, appearing earlier in girls than in boys [57]. Girls may try to hide these changes in PA, considering them unattractive [33,58], negatively impacting their self-perception of PF.

Another objective of the present research was to study whether there were differences in the self-perception of PF according to the students’ school location. This study shows that students from rural schools showed slightly higher self-perceived fitness scores in most physical abilities than students from urban schools; however, these differences were not significant. Therefore, we could not affirm that the location of the center influences self-perception of PF. There is not much literature that analyzes the physical perception of adolescents as a function of the location of the educational center. However, as mentioned above, the level of PA [30,51] and several studies have reported that young people in rural locations engage in a greater amount of PA than young people in urban locations [59,60], and this level of PA may influence their self-perceptions of PF. In this regard, several studies report that adolescents from rural populations report higher health-related quality of life, higher sleep quality and greater psychological well-being related to school environment and autonomy. These data may be related to the greater likelihood of urban adolescents to be influenced by physical factors such as pollution, less access to nature, high population density, and social factors such as fast-paced life and stress [61,62,63,64].

As for the correlations between self-perceived PF with BMI and age, the correlations between BMI and self-perceived PF were inverse, medium and significant for the items of overall physical fitness (r = −0.202), cardiorespiratory endurance (r = −0.226) and movement speed (r = −0.268). Therefore, according to these data, it could be affirmed that the lower the BMI, the better the self-perception of general PF. Other studies support these results to some extent, showing a strong inverse association between PF and overweight in adolescents [37,39], and a lower performance in PF in overweight and obese adolescents compared to those with a normal weight [37,65,66]. The same happens with the level of PA, where it has been observed in several studies that the higher the weight, the lower the level of PA in adolescents [30,67,68]. The lower level of PF and lower level of PA associated with overweight and obesity seems to influence self-esteem and perceptions related to body satisfaction, with overweight adolescents with a lower level of PA having more negative perceptions associated with physical perception and lower self-esteem [69,70,71]. The correlations between age and self-perceived PF were significant and inverse for the items referring to global muscular strength, movement speed and cardiorespiratory endurance; however, these correlations were low according to Mondragón Barrera [43] with the exception of cardiorespiratory endurance (r = −0.138), where the correlations were medium.

### 4.1. Practical Implications

One of the main findings of the present research is that adolescent girls show a less positive self-perception of PF than adolescent boys. This lower self-perception of PF by girls could be related to the tendency of girls to decrease their PA level or directly quit sports practice [72] due to lack of time for practice [73], body image, physical–social anxiety and also due to fatigue/laziness [74]. Lack of motivation to exercise and the stress of educational tasks in subjects other than physical education make PA less of a priority on a daily basis [75]. Therefore, in the case of physical education, public administrations and education departments should emphasize continuous teacher training, using methodologies and designing learning situations that address the interests, motivations and expectations of students. It would be interesting to advocate for equal opportunities and ensuring that students feel competent in their learning and are interested in the practice of PA outside of school hours.

In this research, it was also observed that a lower self-perception of PF by adolescents was related to a higher BMI for most of the self-reported physical abilities. This inverse association between PF and overweight in adolescents [34,36] with the level of PA has been observed in several studies where the higher the weight, the lower the level of PA in adolescents [30,64,65]. Therefore, the departments of education should advocate the fight against sedentary lifestyles and overweight among students, supporting educational plans that involve active methodologies, from the point of view of PA. These include the development of active classes in subjects other than physical education, active breaks between classes and use of other methods in which PA is included in the teaching–learning process of students.

### 4.2. Limitations

This was a cross-sectional study; therefore, it was not possible to establish cause–effect relationships. In future research, it would be enriching to explore these results in greater depth in order to establish possible causal relationships.

The participants in the present study are students from schools in the Spanish autonomous community of Extremadura, and the sociocultural variables of this community may have influenced the results obtained. Therefore, it would be interesting to develop this type of study in more regions of Spain and to be able to compare attitudes in other regions of Spain or extrapolate it to other countries in Europe and the world.

In future research, studying self-perceived fitness relationships by cycle would be novel to analyze the evolution of self-perceived fitness over the years.

## 5. Conclusions

It can be concluded that gender, age and BMI influenced self-perception of FP, but not the school environment. Therefore, in order to motivate students, especially girls, it would be interesting for educational administrations to promote initiatives to encourage PA through equal opportunities, improving their FP and enabling them to lead a healthier lifestyle, providing continuous training tools for teachers in order to make students competent, and implementing strategies such as the development of active classes in subjects other than physical education, active breaks between classes and the use of other methods in which PA is included in the teaching–learning process of students. In other words, it should promote initiatives that advocate the fight against sedentary lifestyles, excess weight and future chronic diseases.

## Figures and Tables

**Table 1 children-10-00604-t001:** Sample characterization (N = 961).

Variable	Categories	N	%
Sex	Boy	564	48.8
	Girl	591	51.2
Level	1º CSE	221	19.1
	2º CSE	213	18.4
	3º CSE	166	14.4
	4º CSE	277	24
	1º Baccalaureate	247	21.4
	2º Baccalaureate	21	2.7
Center location	Rural	368	31.9
	Urban	787	68.1
**Variable**		M	SD
Age		14.71	1.58
BMI		20.61	3.36

N: number; %: percentage; SD: standard deviation; M: Mean; CSE: Compulsory Secondary Education; BMI: Body Mass Index.

**Table 2 children-10-00604-t002:** Scores and differences obtained according to sex and center location for the items of the FP VAS A scale.

Items	Sex				Center Location			
	Men	Women			Rural	Urban		
	M (SD)	M (SD)	*p*	g	M (SD)	M (SD)	*p*	*g*
1. My overall physical fitness is	7.30 (1.75)	6.47 (2.07)	<0.001 *	0.431	6.99 (1.79)	6.82 (2.04)	0.466	0.086
2. My cardiorespiratory endurance (ability to do physical activities for a long time) is:	7.32 (2.13)	6.32 (2.41)	<0.001 *	0.438	6.82 (2.23)	6.80 (2.37)	0.844	0.009 *
3. My overall muscle strength is:	7.17 (1.81)	6.05 (2.16)	<0.001 *	0.560	6.69 (2.03)	6.55 (2.09)	0.245	0.067
4. My movement speed (the ability to run very fast) is:	7.68 (1.93)	6.30 (2.30)	<0.001 *	0.647	6.97 (2.13)	6.97 (2.28)	0.721	0.001 *
5. My overall flexibility is:	5.47 (2.43)	6.35 (2.40)	<0.001 *	0.3363	5.90 (2.44)	5.93 (2.46)	0.792	0.001 *
Visual Analogue Fitness Perception Scale for Adolescents (FP VAS A)	6.98 (1.43)	6.29 (1.69)	<0.001 *	0.438	6.67 (1.52)	6.61 (1.64)	0.669	0.037 *

Note: *p* < 0.05 is significant *. M = mean value; SD = Standard deviation. Each score from the VAS PFA is based on a Likert scale (1–10).

**Table 3 children-10-00604-t003:** Correlation between FP VAS A scale items with age and BMI.

Items	Age *ρ* (*p*)	IMC *ρ* (*p*)
1. My overall physical fitness is	−0.041 (0.162)	−0.202 (<0.001)
2. My cardiorespiratory endurance (ability to do physical activities for a long time) is:	−0.138 (<0.001)	−0.226 (<0.001)
3. My overall muscle strength is:	−0.062 (0.035)	0.043 (0.148)
4. My movement speed (the ability to run very fast) is:	−0.076 (0.009)	−0.268 (<0.001)
5. My overall flexibility is:	0.023 (0.435)	−0.063 (0.033)
Visual Analogue Fitness Perception Scale for Adolescents (FP VAS A)	−0.092 (0.002)	−0.197 (<0.001)

The correlation is significant at *p* < 0.005; Each score from the VAS PFA is based on a Likert scale (1–10).

## Data Availability

The datasets are available through the corresponding author on reasonable request.

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
