# Peer review of "Analysis of Self-Perceived Physical Fitness of Physical Education Students in Public Schools in Extremadura (Spain)"

_children, 2023, doi:10.3390/children10030604_

Round 1

Reviewer 1 Report

This is a cross-sectional study with the objective of (1) To analyze, with the Visual Analogue Fitness Perception Scale for Adolescents (FP VAS A), the self-perceived physical fitness (PF) of high school students, (2) to investigate if there are differences according to sex and school location, (3) to study the correlations between the items of the FP VAS A with age and body mass index (BMI).

The study is well designed; the results are clear and achieve the objectives. I send a few suggestions:

minor revisions

Insert the study design (cross-sectional) in the abstract and methods.

Line 22 - "51.2% girls %". Delete the last %.

Lines 108 to 114 and Table 1 refer to results. It is suggested to send it to the results section. BMI information was also missing in the text "and the mean BMI (kg weight/height in meters2"), line 114.

Author Response

REVIEWER 1

Comments and Suggestions for Authors

This is a cross-sectional study with the objective of (1) To analyze, with the Visual Analogue Fitness Perception Scale for Adolescents (FP VAS A), the self-perceived physical fitness (PF) of high school students, (2) to investigate if there are differences according to sex and school location, (3) to study the correlations between the items of the FP VAS A with age and body mass index (BMI).

The study is well designed; the results are clear and achieve the objectives. I send a few suggestions:

Authors’ response: Thank you for your review of our manuscript. We have carefully considered your comments and believe that the quality of the paper has improved after incorporating your suggestions. Below are our responses to your suggestions:

Minor revisions

Insert the study design (cross-sectional) in the abstract and methods.

Authors’ response: Thank you very much, your suggestion has been inserted in the abstract.

Line 22 - "51.2% girls %". Delete the last %.

Authors’ response: It has been removed, thank you very much

Lines 108 to 114 and Table 1 refer to results. It is suggested to send it to the results section. BMI information was also missing in the text "and the mean BMI (kg weight/height in meters2"), line 114.

Authors’ response: Thank you, you are right. We have added it to the results section, adhering to your instructions.

Reviewer 2 Report

A good study looking at self-perceived physical fitness in physical education students in public school systems. It is essential because school age and adolescence are significant in directing an individual's interest in fitness, which is shown to impact how it impacts the lifestyle and things like obesity and chronic diseases such as cardiovascular disease associated with physical fitness in adult years. Giving such studies is therefore crucial to add meaningful data for developing programs aiming at improving physical fitness education and resources, say in schools and growing populations.

The abstract should flow better and in a reader-friendly way.

One central premise of the study is to see the effect of self-reported PF on secondary school students and to investigate whether there are differences depending on center location - rural and urban. Still, looking at the sample size, 31.9% studied in rural schools, and 68.1% were surveyed in urban schools; what are measures taken to make sure the comparison between urban and rural was fair and statically apt to make the correlation?

Also, did the authors look at the wealth disparity and lifestyle differences like eating habits among the participants in urban and rural settings? Because from some studies, we know that wealth disparity and family income shows an effect on such studies 

The results section could be better for a more detailed explanation of the findings.

The discussion section is good, and I applaud the authors for including a limitations and practical implications section in the manuscript.

It could be modified to make the text flow better, particularly in the methods and results sections.

Author Response

REVIEWER 2

Comments and Suggestions for Authors

A good study looking at self-perceived physical fitness in physical education students in public school systems. It is essential because school age and adolescence are significant in directing an individual's interest in fitness, which is shown to impact how it impacts the lifestyle and things like obesity and chronic diseases such as cardiovascular disease associated with physical fitness in adult years. Giving such studies is therefore crucial to add meaningful data for developing programs aiming at improving physical fitness education and resources, say in schools and growing populations.

Authors’ response: Thank you for your review of our manuscript. We have carefully considered your comments and believe that the quality of the paper has improved after incorporating your suggestions. Below are our responses to your suggestions:

The abstract should flow better and in a reader-friendly way.

Authors’ response: Thank you very much, we have modified it according to your suggestion. We believe that the abstract is now more fluid.

One central premise of the study is to see the effect of self-reported PF on secondary school students and to investigate whether there are differences depending on center location - rural and urban. Still, looking at the sample size, 31.9% studied in rural schools, and 68.1% were surveyed in urban schools; what are measures taken to make sure the comparison between urban and rural was fair and statically apt to make the correlation?

Authors’ response: Thank you very much, by using a convenience sampling method and selecting as indicated in the procedure the contact of all schools in Extremadura, we tried to balance the sample even though the demographics of Extremadura indicate that there are more children in urban areas than in rural areas.

Also, did the authors look at the wealth disparity and lifestyle differences like eating habits among the participants in urban and rural settings? Because from some studies, we know that wealth disparity and family income shows an effect on such studies 

Authors’ response: You are right, for further studies we will include these variables you allude to, but it is difficult to obtain some information especially from children. Anyway, thank you very much, we will keep it in mind for future lines.

The results section could be better for a more detailed explanation of the findings.

Authors’ response: Thank you, a more detailed explanation of the results has been made.

The discussion section is good, and I applaud the authors for including a limitations and practical implications section in the manuscript.

Authors’ response: Thank you very much for your appreciation. Special attention has been given to this section because we consider it relevant for the scientific community and to bring science closer to the public.

It could be modified to make the text flow better, particularly in the methods and results sections.

Authors’ response: Thank you for your suggestions, we have tried to make it smoother and now the article has been improved thanks to you.

Reviewer 3 Report

Abstract

- Conclusion of the abstract is just a replication of the results. Please add some sentences as conclusion or practical implications.

Introduction

- Although the rational of the stud is clearly stated, the authors did not provide scientific documentations on the relationships between physical fitness, age, and BMI. In fact, few literatures have been reported for these variables. Please consider this issue in the introduction.

- Why did you consider center location as a independent variable? Please explain in the introduction.  

Method

- Regarding the participates, how did you choose this number of sample? Please add info using G*Power for the sample size. Also, was there a drop in the participants due to not completing the protocol? Please explain.

- Regarding the FP VAS A, may you please add some sample items? As well, what about the validity and reliability of the instrument? Please explain.

Results

- Table 2: please use boys and girls instead of men and women.

- Table 3: What is IMC????

- Please remove the data of reliability of the instrument and transfer it into the method.

- Is it possible to use Regression Analysis for prediction of self-perception PF from age and BMI? It yes, please add the results.

Discussion & Conclusion

- Discussion is well written. However, conclusion is just a replication of the results. Please reconsider conclusion.

Good luck

Author Response

REVIEWER 3

Comments and Suggestions for Authors

Authors’ response: Thank you for your review of our manuscript. We have carefully considered your comments and believe that the quality of the paper has improved after incorporating your suggestions. Below are our responses to your suggestions:

Abstract

- Conclusion of the abstract is just a replication of the results. Please add some sentences as conclusion or practical implications.

Authors’ response: Thank you very much for the suggestion. We have added practical implications.

Introduction

- Although the rational of the stud is clearly stated, the authors did not provide scientific documentations on the relationships between physical fitness, age, and BMI. In fact, few literatures have been reported for these variables. Please consider this issue in the introduction.

- Why did you consider center location as a independent variable? Please explain in the introduction. 

Authors’ response: Thank you for your enquiry. There is some controversy about the influence of the school environment. It has always been reported that people in rural schools had better FP, but lately, due to the increasing diversity of extracurricular activities, it seems that it is the urban population that has better FP. We have added a paragraph in the introduction with their references. Sinceraly, thanks.

Method

- Regarding the participates, how did you choose this number of sample? Please add info using G*Power for the sample size. Also, was there a drop in the participants due to not completing the protocol? Please explain.

Authors’ response: Thank you very much, no sample calculation was performed since the Junta de Extremadura did not provide us with the N. Therefore, the convenience method explained in the manuscript was used in which all the educational centers were contacted in order to obtain the largest possible sample from those centers that wished to collaborate with the study. There was no reduction in the sample since the study was carried out with those classes with which all the informed consents accepted were approved.

- Regarding the FP VAS A, may you please add some sample items? As well, what about the validity and reliability of the instrument? Please explain.

Authors’ response: Thank you very much, all the items of the instrument are explained in table 2. Thank you very much, the cronbach's alpha coefficient has been included in the instruments section.

Results

- Table 2: please use boys and girls instead of men and women.

Authors’ response: Thanks, it has been modified.

- Table 3: What is IMC????

Authors’ response: Thanks, it’s in Spanish language. Sorry it was a mistake, already it was modificied.

- Please remove the data of reliability of the instrument and transfer it into the method.

Authors’ response: Thank you very much, we have done so.

- Is it possible to use Regression Analysis for prediction of self-perception PF from age and BMI? It yes, please add the results.

Authors’ response: Thank you very much, we have not included the regression analysis since less than 1% is predicted. Thank you very much, we have not included the regression analysis since less than 1% is predicted.

Discussion & Conclusion

- Discussion is well written. However, conclusion is just a replication of the results. Please reconsider conclusion.

Authors’ response: Thank you very much we have adhered to your suggestions with the conclusions. The manuscript has been greatly improved thanks to all your contributions.

Good luck

Round 2

Reviewer 3 Report

Thank you for your revision. I am satisfied with the revised version of the paper. My only comment is that conclusion of the abstract is not edited based on my comment. It is just a replication of the results. Please replace it with a practical implication. Thanks 

Author Response

Thank you for your revision. I am satisfied with the revised version of the paper. My only comment is that conclusion of the abstract is not edited based on my comment. It is just a replication of the results. Please replace it with a practical implication. Thanks 

Authors’ response: Thank you for your words toward our manuscript. Adhering to their suggestions, we have modified the conclusions by adding practical implications. Thank you for everything.
